# Assessing the capacity of symptom scores to predict COVID-19 positivity in Nigeria: a national derivation and validation cohort study

Kelly Osezele Elimian ,[1,2] Olaolu Aderinola,[1] Jack Gibson,[3] Puja Myles,[3] Chinwe Lucia Ochu [1] Carina King,[2] Tochi Okwor,[1] Giulia Gaudenzi [2] Adebola Olayinka,[4] Habib Garba Zaiyad,[5] Cornelius Ohonsi,[1] Blessing Ebhodaghe,[1] Chioma Dan-Nwafor,[1] William Nwachukwu,[1] Ismail Adeshina Abdus-salam,[6] Oluwatosin Wuraola Akande [1,7] Olanrewaju Falodun,[8] Chinedu Arinze,[1] Chidiebere Ezeokafor,[9] Abubakar Jafiya,[1] Anastacia Ojimba,[10] John Tunde Aremu,[11] Emmanuel Joseph,[12] Abimbola Bowale,[13] Bamidele Mutiu,[14] Babatunde Saka,[15] Arisekola Jinadu,[1] Khadeejah Hamza,[16] Christian Ibeh,[17] Shaibu Bello,[18] Michael Asuzu,[19] Nwando Mba,[1] John Oladejo,[1] Elsie Ilori,[1] Tobias Alfvén,[2] Ehimario Igumbor,[1,20] Chikwe Ihekweazu[1]

**Correspondence to**
Dr Kelly Osezele Elimian;
kelly.elimian@ki.se

## ABSTRACT

**Objectives** This study aimed to develop and validate a symptom prediction tool for COVID-19 test positivity in Nigeria.

**Design** Predictive modelling study.

**Setting** All Nigeria States and the Federal Capital Territory.

**Participants** A cohort of 43 221 individuals within the national COVID-19 surveillance dataset from 27 February to 27 August 2020. Complete dataset was randomly split into two equal halves: derivation and validation datasets. Using the derivation dataset (n=21 477), backward multivariable logistic regression approach was used to identify symptoms positively associated with COVID-19 positivity (by real-time PCR) in children (≤17 years), adults (18–64 years) and elderly (≥65 years) patients separately.

**Outcome measures** Weighted statistical and clinical scores based on beta regression coefficients and clinicians' judgements, respectively. Using the validation dataset (n=21 744), area under the receiver operating characteristic curve (AUROC) values were used to assess the predictive capacity of individual symptoms, unweighted score and the two weighted scores.

**Results** Overall, 27.6% of children (4415/15 988), 34.6% of adults (9154/26 441) and 40.0% of elderly (317/792) that had been tested were positive for COVID-19. Best individual symptom predictor of COVID-19 positivity was loss of smell in children (AUROC 0.56, 95% CI 0.55 to 0.56), either fever or cough in adults (AUROC 0.57, 95% CI 0.56 to 0.58) and difficulty in breathing in the elderly (AUROC 0.53, 95% CI 0.48 to 0.58) patients. In children, adults and the elderly patients, all scoring approaches showed similar predictive performance.

**Conclusions** The predictive capacity of various symptom scores for COVID-19 positivity was poor overall. However, the findings could serve as an advocacy tool for more

## STRENGTHS AND LIMITATIONS OF THIS STUDY

⇒ This study provides the early evidence on the predictive capacity of symptom scores for COVID-19 positivity in a sub-Saharan African country.

⇒ The study used the train-test split-sample method to randomly split the analysed dataset into two (ie, training and testing datasets), thereby allowing for validation of the generated clinical scores.

⇒ The study adopted a participatory approach in the derivation of clinically weighted score by engaging clinicians managing patients with COVID-19, thereby enhancing the clinical relevance of the findings.

⇒ Splitting the dataset for both derivation and validation of symptom scores might have underestimated the predictive performance of the models due to loss of power.

⇒ The study lacked sufficient clinical laboratory data that have been shown to be significant in designing COVID-19 diagnostic models.

investments in resources for capacity strengthening of molecular testing for COVID-19 in Nigeria.

## INTRODUCTION

The index case of COVID-19 in Nigeria was recorded on 27 February 2020, in Ogun State; one of the contacts of the index case was later diagnosed with the disease in the same state. As of epidemiological week 44 (26 October–1 November), Nigeria had recorded 62 964 confirmed cases, of whom 1146 died and 58 790 recovered, resulting in a case fatality rate of 1.8%.[1] The rapid increase in

BMJ

COVID-19 cases in Nigeria caused a strain on the healthcare systems, particularly in the area of molecular testing using real-time PCR (RT-PCR). Deficiencies in molecular laboratory capacity can hinder the rapid identification of persons with COVID-19 infection and initiation of appropriate treatment and public health measures.[2] For example, as of 9 April 2020, only 5000 tests for SARS-CoV-2 were performed across Nigeria by eight laboratories across six states.[3] Concerted efforts have since been made to expand molecular testing capacity across the country, with 673 183 tests for SARS-CoV-2 conducted and the Nigeria Centre for Disease Control (NCDC) COVID-19 laboratory networks expanded to 69 functioning laboratories as of 6 November 2020.[4] However, with a population of over 200 million and community transmission of COVID-19, the testing rate for Nigeria to date is still well below the Africa Centre for Disease Control-set target of 1% for population. Generally, laboratory capacity to test for SARS-CoV-2 remains a challenge in many sub-Saharan African countries, especially in the early phase of the COVID-19 pandemic.[5]

Although PCR testing is considered the gold standard for COVID-19 diagnosis, limited access and centralised laboratory systems often contribute to delay.[6] This has contributed to making diagnostic results unavailable for urgent decision-making by frontline healthcare workers, with consequent effect on healthcare delivery and increased risk of nosocomial infection.[7] Between 27 February and 6 June 2020, for example, the median turnaround time for laboratory diagnosis of COVID-19 in Nigeria was 2 (IQR 1–4) days; 9% of the 12 289 confirmed COVID-19 cases during this period was recorded among healthcare workers.[8] Thus, to mitigate the impact of COVID-19 on the healthcare system, while also improving community surveillance and care for patients, rapid and accurate diagnosis are crucial.[9] NCDC and its technical partners are currently conducting a study to validate rapid diagnostic test kits to determine their accuracy and clinical utility in Nigeria.

One approach to supplement laboratory testing is, however, the use of diagnostic prediction models that use clinical variables (including standard blood counts when available) to estimate the likelihood of individuals being COVID-19 positive or experiencing poor clinical outcome.[9] A systematic review of developed and/or validated prediction models found the most frequently reported predictors of COVID-19 diagnosis to be influenza-like symptoms (eg, chills, fatigue) and neutrophil count, with C-statistic estimates ≥0.90 (an excellent discriminative performance).[9] However, to date, these prediction models have been judged to be at a high risk of bias, largely because they were fitted on data that were not representative of the target population and have suffered from poor reporting on intended use of the models or their calibration performance. While the reported performances of models to date are encouraging, their adoption in clinical practice has not been recommended.[9] In addition, there has been a dearth of evidence from sub-Saharan Africa as the majority of available models use data from China and European countries, thus limiting their generalisability to settings such as Nigeria with different COVID-19 epidemiology, healthcare systems, socioeconomic conditions and health-seeking behaviours.

The majority of confirmed COVID-19 cases in Nigeria are asymptomatic at the point of diagnosis[8]; for the confirmed cases who present with symptoms, the most common symptoms tend to be non-specific (eg, fever, cough and difficulty in breathing).[8 10 11] Using data from 27 February to 6 June 2020, presentation with cough, fever, loss of smell and loss of taste was found to be positively and independently associated with COVID-19 positivity in Nigeria.[12] But the dearth of evidence on the predictive capacity of clinical symptoms to predict COVID-19 in Nigeria (and indeed in sub-Saharan Africa) is the rationale for this study. Thus, we aimed to develop and validate the predictive capacity of clinical signs and symptoms with regard to testing positive for COVID-19 and to investigate whether there are any gains in the predictive capacity of statistically and clinically derived weighted combined symptom scores, as compared with an unweighted combined score in Nigeria.

## METHODS

### Study design and data source

This is a predictive modelling study using a retrospective cohort of persons enrolled in the Surveillance, Outbreak Response Management and Analysis System (SORMAS) database from 27 February to 27 August 2020. A detailed description of SORMAS in the COVID-19 context in Nigeria is available elsewhere.[8] Briefly, SORMAS is an open-source real-time electronic health surveillance and laboratory database, which has been in use in Nigeria since 2017. SORMAS is hosted and coordinated at the NCDC's headquarter in Abuja.

### Study participants

The study population comprised children (≤17 years), adults (18–64 years) and the elderly (≥65 years) who were tested for SARS-CoV-2 infection. Eligibility for RT-PCR test during the study period was based on the NCDC COVID-19 standard case definition for suspected cases (online supplemental table 1).[13] However, provisions were made to test persons concerned about COVID-19 infection on presentation to designated testing centres regardless of whether they met the NCDC case definitions or not.

### Data collection and management

Sociodemographic (eg, age, sex, occupation, education and geopolitical zone of residence) and clinical presentation (signs and symptoms in the 14 days prior to testing) were collected from all persons tested. The collection and transportation of respiratory samples (oropharyngeal and nasopharyngeal swabs) for laboratory analysis were facilitated by trained healthcare workers according to the

NCDC guidelines.[14] Testing for SARS-CoV-2 by RT-PCR was performed according with the WHO guidelines.[15] All the collected data across the country, including laboratory diagnostic outcomes, were submitted by trained healthcare workers to NCDC via the SORMAS application installed on tablets or laptops. Patients who tested positive for COVID-19 were managed either in a government health facility or at home, depending on the illness severity or availability of bed space at health facilities, while observing the NCDC case management guidelines.[14]

Since this was an analysis of a secondary dataset, there was no formal sample size calculation for this study; however, the study met the standard sample size requirement of 10 outcome events per degree of freedom in prediction models (eg, 10 binary variables in the model require 100 COVID-19 positive cases). We used the train-test split-sample method to randomly split the complete dataset into two (ie, training and testing datasets), assigning half of the records to either the training (derivation) or testing (validation) datasets based on a random number generated within the statistical software. However, we made four major assumptions to define the study eligibility criteria. First, we assumed that a missing variable was indicative of absence (ie, record not present), supported by findings (p<0.001) from a $\chi^2$ test of association between two ad hoc variables 'any-missing' and 'any-absent'. Second, we decided that eligible study

participants needed to have complete records for both age and sex since these are crucial demographic variables. Third, given the high proportion of asymptomatic COVID-19 cases in Nigeria,[8] the study participants needed to have had at least one symptom positively recorded (ie, symptomatic). Lastly, analyses were performed separately for children, adults and the elderly, given the evidence supporting the age dependence of COVID-19 symptom presentation, with children more likely to be asymptomatic than adults.[16 17] Figure 1 shows a flowchart showing the processes for selecting the study population as stated in the eligibility criteria (definitions of suspected and probable cases are available in the NCDC case definition[13]). All symptoms were coded as '1' if present and '0' if not. Sex was classified as a binary variable, with '1' and '0' indicative of male and female gender, respectively.

## Outcome and predictor variables

The outcome was COVID-19 positivity, defined as either presence or absence of SARS-CoV-2 by RT-PCR confirmation (yes/no). Clinical prediction variables were informed by evidence from our previous study using the same dataset, although it covered a shorter period (27 February to 5 June 2020).[12] The clinical variables were collected using a combination of self-reports by COVID-19 suspected cases and/or their caretakers as well as by objective assessment by a healthcare worker. The

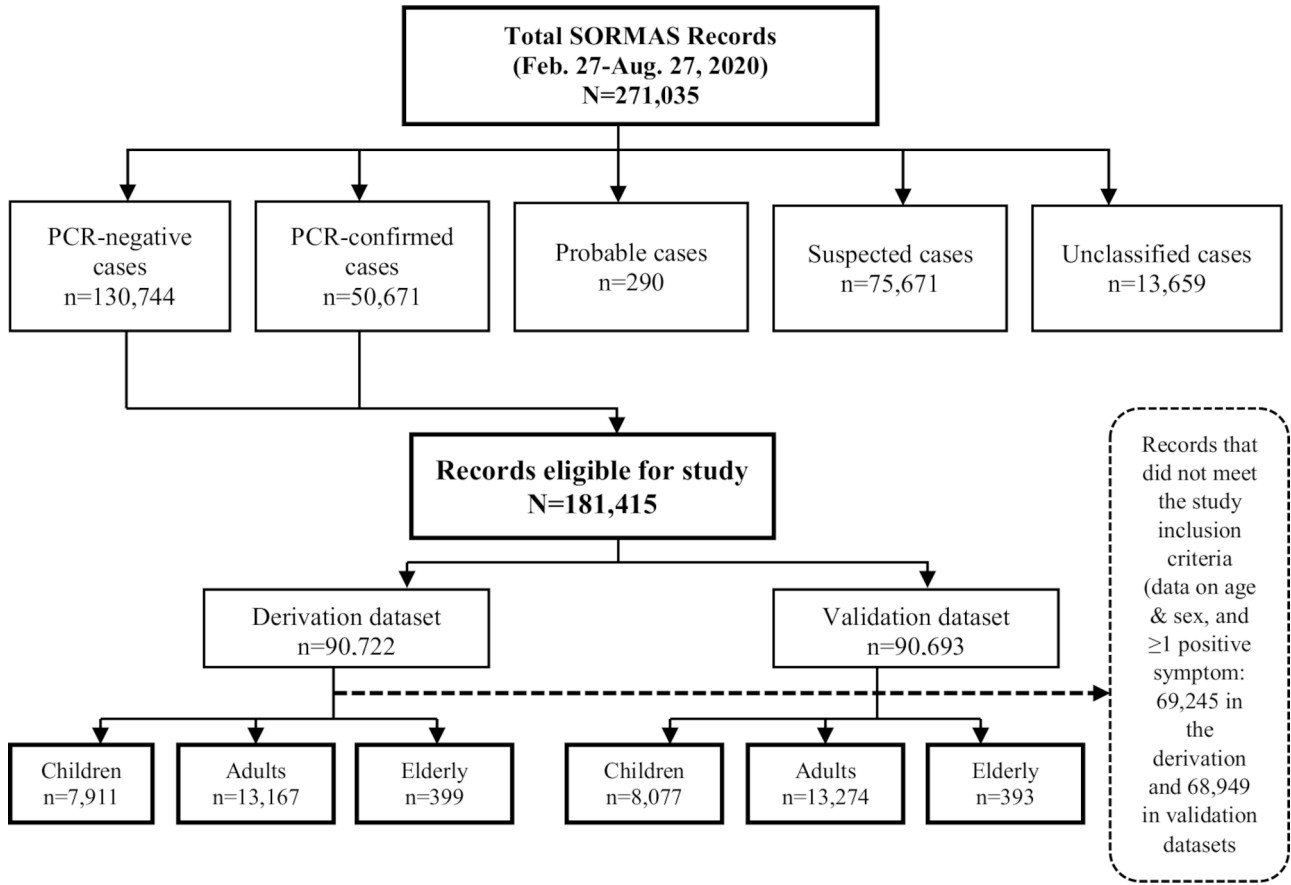

**Figure 1** Flowchart showing the processes for selecting records from the Surveillance, Outbreak Response Management and Analysis System.

three categories of clinical signs and symptoms (each coded yes/no) with their respective examples include (1) 'obvious or as observed by a healthcare worker' (chills/sweat, cough, breathing difficulty, rapid breathing and runny nose); (2) 'elicitable or information volunteered or given on inquiry by patients/relatives' (abdominal pain/diarrhoea, gastrointestinal tract symptoms, chest pain, fatigue/weakness, headache, musculoskeletal pain, sore throat, loss of taste and loss of smell); (3) and 'measurable as assessed by a healthcare worker' (fever, defined as an axillary temperature ≥37.5°C).

## Statistical analyses

The data were set up and analysed using Stata V.13.0 (Stata Corporation, College Station, Texas, USA).

### Model development

Using the derivation dataset, the association between individual symptoms and COVID-19 positivity was investigated, in turn, using univariable logistic regression. Analyses were stratified by children, adults and the elderly. The results are presented as unadjusted ORs and 95% CIs. To determine which of the symptoms were independently associated with COVID-19 positivity, we first conducted a test for multicollinearity by the various age groups (children, adults and the elderly) using the variance inflation factor (VIF) and tolerance values of each statistically significant symptom identified from the univariable logistic regression model (see online supplemental table 2a, b and c for results). This was to ensure that symptoms were independent of each other or not collinear and therefore suitable for combining into a single score. Both VIF and tolerance values measure how much the regression coefficient for a symptom is determined by the other symptoms in the model; low VIF and high tolerance values suggest the absence of multicollinearity and vice versa.[18] Thereafter, we used the backward multivariable logistic regression approach to select all the statistically significant symptoms for the model. As before, analyses were conducted for children, adults and the elderly separately.

We then created three different symptom scores. First, an unweighted combined score was calculated for each person (separately for children, adults and the elderly) by allocating '1' point for each symptom before summation. The unweighted combined scores were included in the model as a continuous variable. The combined scores were further recoded as binary categorical variables to represent clinical scores ≥1 (yes/no), ≥2 (yes/no), ≥3 (yes/no) and ≥4 (yes/no). Additionally, weighted combined scores were developed using two approaches. First, we used the beta regression coefficients obtained from the multivariable logistic regression model that was run previously. Each symptom was given the value of '1' if present and '0' if otherwise. Each symptom was then multiplied by its weight, obtained by multiplying the respective beta regression coefficients (or log of the ORs) by 10. Essentially, the beta regression coefficients measure the relative prognostic strength of each symptom

when they are included simultaneously in a multivariable regression model so that the bigger the value of a symptom, the more its weight. Second, we engaged six clinicians (including infectious disease consultants) who have been managing patients with COVID-19 in treatment centres and at home in the Federal Capital Territory, Gombe, Delta and Kaduna. Here, the clinicians were provided with a list of symptoms identified to be statistically significant from the univariable logistic models, with the freedom to add any missing sign or symptom in the model as needed. Each clinician was then asked to independently assign a weight of 1–5 for each symptom based on experiences from managing patients with COVID-19 (see online supplemental table 3 for detail of weighted scores assigned by each clinician for the identified symptoms). Similar to the statistical weighting approach, each symptom was multiplied by the average of the combined weights assigned by all the clinicians. Predictive capacity of the various score thresholds was also compared to see which combination of symptoms was more predictive of COVID-19 positivity.

### Model validation

The different models were then applied to the validation dataset. Separately for children, adults and the elderly, predictive capacity was assessed in terms of sensitivity, specificity, area under the receiver operating characteristic (ROC) curve (AUROC) value, positive predictive value (PPV) and negative predictive value (NPV)[19]; and ROC curves were plotted for the comparison of clinical prediction scores.

Where applicable, this study is reported according to the TRIPOD (Transparent Reporting of a multivariable prediction model for Individual Prognosis or Diagnosis) statement.[20]

### External calibration belt and test for model goodness of fit

Using the validation dataset, external calibration test and belt were used to evaluate the goodness of fit of our models, an approach that examines the relationship between estimated probabilities and observed outcome rates.[21] Tests and belts often return concordant outputs: non-significant tests are often associated with the belt encompassing the 45 degree lines (good fit) and significant tests with the belt deviating from the bisector (poor fit).

### Patient and public involvement

Being an analysis of deidentified secondary dataset, it was not possible to involve patients or the public in the design, or conduct, or reporting or dissemination plans of this study. However, clinicians who have been managing patients with COVID-19 both in health facilities and at home in Nigeria were actively engaged both in planning the study and in developing the clinical scores.

## RESULTS

### Description of study population

The baseline sociodemographic and clinical characteristics of the study participants in the combined datasets are presented in table 1. Overall, 15 988 children met the study eligibility criteria, 4415 (27.6%) of whom tested positive for COVID-19 and 62.2% were male. A total of 26 441 adults met the study eligibility criteria, 9154 (34.6%) of whom tested positive for COVID-19, 64.0% (16 819/26 441) were male, and 11.3% (2977/26 441) were healthcare workers. Seven hundred and ninety-two elderly patients met the study eligibility criteria, 317 (40.0%) of whom tested positive for COVID-19 and 62.3% were male. Similar characteristics were observed in the derivation (online supplemental table 4) and validation (online supplemental table 5) cohorts for all age groups.

### Performance of individual symptoms for predicting COVID-19 positivity

In the unadjusted model, presentation with cough, runny nose, fatigue, loss of taste, loss of smell and fever was significantly associated with higher odds of COVID-19 positivity in children (left panel of table 2). In the adjusted model, however, presentation with the following symptoms remained significantly (p<0.001) associated with higher odds of COVID-19 positivity: cough (adjusted OR (aOR) 1.32, 95% CI 1.19 to 1.47), runny nose (aOR 1.48, 95% CI 1.32 to 1.66), fatigue (aOR 1.53, 95% CI 1.21 to 1.93), loss of taste (aOR 2.26, 95% CI 1.63 to 3.12), loss of smell (aOR 4.87, 95% CI 3.66 to 6.49) or fever (aOR 1.47, 95% CI 1.33 to 1.63). Regarding predictive performance, loss of smell recorded the highest AUROC value (0.56, 95% CI 0.55 to 0.56). Specifically, cough recorded the highest sensitivity (47.7%, 95% CI 45.6% to 49.7%) and NPV (74.7%, 95% CI 73.4% to 75.9%), while loss of smell recorded the highest specificity (98.1%, 95% CI 97.7% to 98.4%) and PPV (72.9%, 95% CI 68.3% to 77.2%) (see details in online supplemental table 6).

The presentation of symptoms significantly associated with increased COVID-19 positivity was higher in adults than in children (middle panel of table 2). In the adjusted model, however, it was only the presentation with cough (aOR 1.77, 95% CI 1.64 to 1.91), runny nose (aOR 1.18, 95% CI 1.08 to 1.30), chest pain (aOR 1.50, 95% CI 1.21 to 1.85), fatigue (aOR 1.39, 95% CI 1.17 to 1.64), headache (aOR 1.24, 95% CI 1.11 to 1.39), loss of taste (aOR 2.33, 95% CI 1.83 to 2.96), loss of smell (aOR 4.18, 95% CI 3.30 to 5.28) or fever (aOR 1.58, 95% CI 1.47 to 1.71) that remained independently associated with COVID-19 positivity. Regarding predictive performance, presentation with either fever or cough appeared to be the best predictor of COVID-19 positivity (AUROC 0.57, 95% CI 0.56 to 0.58). Additionally, presenting with cough or fever recorded the highest sensitivity at 54.3% (52.9% to 55.8%) and NPV at 70.8% (69.7% to 71.8%); however, presenting with loss of smell recorded the highest specificity (98.5%; 95% CI 98.2% to 98.8%) and PPV (79.4%, 95% CI 76.0% to 82.5%) (see details in online supplemental table 7).

Only three symptoms in the elderly patients were significantly associated with COVID-19 positivity both in the unadjusted and adjusted models, with cough (aOR 1.59, 95% CI 1.04 to 2.43; p=0.033), difficulty in breathing (aOR 1.74, 95% CI 1.12 to 2.72; p=0.015) or loss of smell (aOR 7.15, 95% CI 1.44 to 35.44; p=0.016) being independently associated with COVID-19 positivity in the latter model (right panel of table 2). Regarding predictive performance, presenting with difficulty in breathing appeared to be the best predictor of COVID-19 positivity (AUROC 0.53, 0.48 to 0.58). However, presenting with cough recorded the highest sensitivity (61.4%, 95% CI 53.3% to 69.0%) and NPV (62.1%, 95% CI 54.1% to 69.6%), while loss of smell recorded the highest specificity (99.1%, 95% CI 97.0% to 99.9%) and PPV (60.0%, 95% CI 14.7% to 94.7%) (see details in online supplemental table 8).

### Performance of unweighted, statistically and clinically weighted scores for predicting COVID-19 positivity

In adults, the statistically derived weighted combined score appeared to be slightly better in predicting COVID-19 positivity (AUROC 0.65), when compared with both the clinically derived weighted combined and unweighted scores with AUROC 0.63 each (table 3). A similar pattern was recorded in children and elderly populations. For individual symptom thresholds, presenting with ≥2 symptoms on the unweighted score appeared to be a better predictor of COVID-19 positivity (AUROC 0.61, 95% CI 0.60 to 0.61) than the other symptom thresholds in adults. For the statistically weighted score, presenting with ≥4 symptoms appeared to be a better predictor of COVID-19 positivity (AUROC 0.59, 95% CI 0.58 to 0.59) than the other symptom thresholds in adults. For the clinically derived weighted score, presenting with either ≥3 or ≥4 symptoms was a better predictor of COVID-19 positivity (AUROC 0.58, 95% CI 0.58 to 0.59) in adults. The detailed results including sensitivity, specificity, PPV and NPV values for each age group are available in online supplemental tables 9-11. Figure 2 shows the ROC curves comparing the predictive performance of unweighted, statistically and clinically derived weighted combined scores in children (A), adult (B) and the elderly (C) populations.

### External calibration belt and test for model goodness-of-fit

The calibration belt in the produced plots and tests for children (p=0.086), adults (p=0.915) and elderly (p=0.091) is presented in figure 3 and suggests that the hypothesis of good calibration is not rejected. The calibration belt and tests for the unweighted, statistically and clinically weighted scores in children, adults and elderly are presented in a supplemental figure.

**Table 1** Baseline sociodemographic and clinical characteristics of all the study participants in relation to COVID-19 infection (combined dataset)

| Variable | Children (≤17 years) | | Adults (18–64 years) | | Elderly (≥65 years) | |
|---|---|---|---|---|---|---|
| | PCR-confirmed cases (n=4415 (%)) | PCR positive and negative cases (n=15 988 (%)) | PCR-confirmed cases (n=9154 (%)) | PCR positive and negative cases (n=26 441 (%)) | PCR-confirmed cases (n=317 (%)) | PCR positive and negative cases (n=792 (%)) |
| **Sociodemographic features** | | | | | | |
| Sex | | | | | | |
| Female | 1773 (40.16) | 6051 (37.85) | 3080 (33.65) | 9622 (36.39) | 110 (34.70) | 299 (37.75) |
| Male | 2642 (59.84) | 9937 (62.15)* | 6074 (66.35) | 16 819 (63.61)* | 207 (65.30) | 493 (62.25)NS† |
| Geopolitical zone‡ | | | | | | |
| South-west | 1302 (29.49) | 3539 (22.14) | 2592 (28.32) | 6245 (23.62) | 86 (27.13) | 185 (23.36) |
| South-south | 1435 (32.50) | 5689 (35.58) | 3228 (35.26) | 10 911 (41.27) | 108 (34.07) | 340 (42.93) |
| South-east | 90 (2.04) | 278 (1.74) | 273 (2.98) | 609 (2.30) | 25 (7.89) | 37 (4.67) |
| North-central | 816 (18.48) | 3056 (19.11) | 1526 (16.67) | 4061 (15.36) | 30 (9.46) | 76 (9.60) |
| North-west | 658 (14.90) | 3118 (19.50) | 1258 (13.72) | 4005 (15.15) | 45 (14.20) | 116 (14.65) |
| North-east | 114 (2.58) | 308 (1.93)* | 277 (3.03) | 610 (2.31)* | 23 (7.26) | 38 (4.80)* |
| Setting | | | | | | |
| Rural | 230 (5.21) | 887 (5.55) | 511 (5.58) | 1562 (5.91) | 35 (11.04) | 74 (9.34) |
| Urban | 2048 (46.39) | 6943 (43.43) | 4487 (49.02) | 13 088 (49.50) | 183 (57.73) | 423 (53.41) |
| Missing | 2137 (48.40) | 8158 (51.03)* | 4156 (45.40) | 11 791 (44.59)NS† | 99 (31.23) | 295 (37.25)§ |
| Education | | | | | | |
| None | 58 (1.31) | 217 (1.36) | 128 (1.40) | 368 (1.39) | 25 (7.89) | 48 (6.06) |
| Nursery | 30 (0.68) | 143 (0.89) | 3 (0.03) | 7 (0.03) | 0 (0.00) | 0 (0.00) |
| Primary | 130 (2.94) | 520 (3.25) | 102 (1.11) | 322 (1.22) | 8 (2.52) | 18 (2.27) |
| Secondary | 385 (8.72) | 1379 (8.63) | 792 (8.65) | 2262 (8.55) | 23 (7.26) | 58 (7.32) |
| Tertiary | 1410 (31.94) | 4066 (25.43) | 3141 (34.31) | 8145 (30.80) | 87 (27.44) | 168 (21.21) |
| Other | 179 (4.05) | 1065 (6.66) | 261 (2.85) | 688 (2.60) | 20 (6.31) | 45 (5.68) |
| Missing | 2223 (50.35) | 8598 (53.78)* | 4727 (51.64) | 14 649 (55.40)* | 154 (48.58) | 455 (57.45)§ |
| Occupation | | | | | | |
| Student/pupil | 599 (13.57) | 2809 (17.57) | 434 (4.74) | 1516 (5.73) | 1 (0.32) | 2 (0.25) |
| Child/housewife | 133 (3.01) | 489 (3.06) | 178 (1.94) | 500 (1.89) | 13 (4.10) | 42 (5.30) |
| Business/trading | 217 (4.92) | 765 (4.78) | 757 (8.27) | 2043 (7.73) | 19 (5.99) | 47 (5.93) |
| Transporter | 11 (0.25) | 80 (0.50) | 45 (0.49) | 174 (0.66) | 0 (0.00) | 3 (0.38) |
| Healthcare worker | 601 (13.61) | 1953 (12.22) | 973 (10.63) | 2977 (11.26) | 7 (2.21) | 20 (2.53) |
| Laboratorian | 15 (0.34) | 40 (0.25) | 26 (0.28) | 63 (0.24) | 0 (0.00) | 0 (0.00) |
| Farmer | 36 (0.82) | 192 (1.20) | 173 (1.89) | 589 (2.23) | 24 (7.57) | 47 (5.93) |
| Animal-related worker | 8 (0.18) | 28 (0.18) | 25 (0.27) | 80 (0.30) | 1 (0.32) | 9 (1.14) |
| Religious/traditional leader | 5 (0.11) | 24 (0.15) | 53 (0.58) | 129 (0.49) | 6 (1.89) | 9 (1.14) |
| Other | 1633 (36.99) | 5126 (32.06) | 4072 (44.48) | 11 408 (43.15) | 164 (51.74) | 375 (47.35) |
| Missing | 1157 (26.21) | 4482 (28.03)* | 2418 (26.41) | 6962 (26.33)* | 82 (25.87) | 238 (30.05)NS† |
| **Clinical signs and symptoms** | | | | | | |
| Clinical outcome | | | | | | |
| Recovered | 2847 (64.48) | 4443 (27.79) | 5694 (62.20) | 7313 (27.66) | 150 (47.32) | 184 (23.23) |
| Dead | 36 (0.82) | 51 (0.32) | 485 (5.30) | 549 (2.08) | 83 (26.18) | 84 (10.61) |
| No outcome yet | 1532 (34.70) | 11 494 (71.89)* | 2975 (32.50) | 18 579 (70.27)* | 84 (26.50) | 524 (66.16)* |
| *Obvious (visible to healthcare workers on sight)* | | | | | | |
| Chills/sweat | | | | | | |

Continued

**Table 1** Continued

| Variable | Children (≤17 years) | | Adults (18–64 years) | | Elderly (≥65 years) | |
|---|---|---|---|---|---|---|
| | PCR-confirmed cases (n=4415 (%)) | PCR positive and negative cases (n=15 988 (%)) | PCR-confirmed cases (n=9154 (%)) | PCR positive and negative cases (n=26 441 (%)) | PCR-confirmed cases (n=317 (%)) | PCR positive and negative cases (n=792 (%)) |
| No | 4355 (98.64) | 15 764 (98.60) | 9009 (98.42) | 26 029 (98.44) | 315 (99.37) | 783 (98.86) |
| Yes | 60 (1.36) | 224 (1.40)NS† | 145 (1.58) | 412 (1.56)NS† | 2 (0.63) | 9 (1.14)NS† |
| Cough | | | | | | |
| No | 2345 (53.11) | 9393 (58.75) | 4130 (45.12) | 14 391 (54.43) | 114 (35.96) | 321 (40.53) |
| Yes | 2070 (46.89) | 6595 (41.25)* | 5024 (54.88) | 12 050 (45.57)* | 203 (64.04) | 471 (59.47)§ |
| Breathing difficulty | | | | | | |
| No | 3936 (89.15) | 14 283 (89.34) | 7549 (82.47) | 22 425 (84.81) | 200 (63.09) | 541 (68.31) |
| Yes | 479 (10.85) | 1705 (10.66)NS† | 1605 (17.53) | 4016 (15.19)* | 117 (36.91) | 251 (31.69)§ |
| Rapid breathing | | | | | | |
| No | 4345 (98.41) | 15 783 (98.72) | 9002 (98.34) | 26 112 (98.76) | 307 (96.85) | 771 (97.35) |
| Yes | 70 (1.59) | 205 (1.28)§ | 152 (1.66) | 329 (1.24)* | 10 (3.15) | 21 (2.65)NS† |
| Runny nose | | | | | | |
| No | 3057 (69.24) | 11 835 (74.02) | 6792 (74.20) | 20 496 (77.52) | 259 (81.70) | 658 (83.08) |
| Yes | 1358 (30.76) | 4153 (25.98)* | 2362 (25.80) | 5945 (22.48)* | 58 (18.30) | 134 (16.92)NS† |
| *Elicitable (can be found out by asking questions of patients/relatives)* | | | | | | |
| Abdominal pain/ diarrhoea | | | | | | |
| No | 4142 (93.82) | 14 939 (93.44) | 8464 (92.46) | 24 580 (92.96) | 297 (93.69) | 728 (91.92) |
| Yes | 273 (6.18) | 1049 (6.56)NS† | 690 (7.54) | 1861 (7.04)NS† | 20 (6.31) | 64 (8.08)NS† |
| GIT symptoms | | | | | | |
| No | 3972 (89.97) | 14 375 (89.91) | 8100 (88.49) | 23 766 (89.88) | 282 (88.96) | 705 (89.02) |
| Yes | 443 (10.03) | 1613 (10.09)NS† | 1054 (11.51) | 2675 (10.12)* | 35 (11.04) | 87 (10.98)NS† |
| Chest pain | | | | | | |
| No | 4286 (97.08) | 15 577 (97.43) | 8782 (95.94) | 25 621 (96.90) | 303 (95.58) | 766 (96.72) |
| Yes | 129 (2.92) | 411 (2.57)NS† | 372 (4.06) | 820 (3.10)* | 14 (4.42) | 26 (3.28)NS† |
| Fatigue | | | | | | |
| No | 4143 (93.84) | 15 229 (95.25) | 8583 (93.76) | 25 014 (94.60) | 292 (92.11) | 750 (94.70) |
| Yes | 272 (6.16) | 759 (4.75)* | 571 (6.24) | 1427 (5.40)* | 25 (7.89) | 42 (5.30)§ |
| Headache | | | | | | |
| No | 3753 (85.01) | 13 712 (85.76) | 7849 (85.74) | 23 283 (88.06) | 287 (90.54) | 738 (93.18) |
| Yes | 662 (14.99) | 2276 (14.24)NS† | 1305 (14.26) | 3158 (11.94)* | 30 (9.46) | 54 (6.82)§ |
| Musculoskeletal pain | | | | | | |
| No | 4299 (97.37) | 15 595 (97.54) | 8901 (97.24) | 25 783 (97.51) | 306 (96.53) | 771 (97.35) |
| Yes | 116 (2.63) | 393 (2.46)NS | 253 (2.76) | 658 (2.49)§ | 11 (3.47) | 21 (2.65)NS† |
| Sore throat | | | | | | |
| No | 3306 (74.88) | 11 968 (74.86) | 6968 (76.12) | 19 583 (74.06) | 259 (81.70) | 649 (81.94) |
| Yes | 1109 (25.12) | 4020 (25.14)NS† | 2186 (23.88) | 6858 (25.94)* | 58 (18.30) | 143 (18.06)NS† |
| Loss of taste | | | | | | |
| No | 3925 (88.90) | 15 283 (95.59) | 8301 (90.68) | 25 268 (95.56) | 303 (95.58) | 770 (97.22) |
| Yes | 490 (11.10) | 705 (4.41)* | 853 (9.32) | 1173 (4.44)* | 14 (4.42) | 22 (2.78)§ |
| Loss of smell | | | | | | |
| No | 3815 (86.41) | 15 160 (94.82) | 8186 (89.43) | 25 198 (95.30) | 307 (96.85) | 778 (98.23) |
| Yes | 600 (13.59) | 828 (5.18)* | 968 (10.57) | 1243 (4.70)* | 10 (3.15) | 14 (1.77)§ |
| *Measurable* | | | | | | |

**Table 1** Continued

| Variable | Children (≤17 years) | | Adults (18–64 years) | | Elderly (≥65 years) | |
|---|---|---|---|---|---|---|
| | PCR-confirmed cases (n=4415 (%)) | PCR positive and negative cases (n=15 988 (%)) | PCR-confirmed cases (n=9154 (%)) | PCR positive and negative cases (n=26 441 (%)) | PCR-confirmed cases (n=317 (%)) | PCR positive and negative cases (n=792 (%)) |
| Fever | | | | | | |
| No | 2350 (53.23) | 9430 (58.98) | 4555 (49.76) | 15 434 (58.37) | 163 (51.42) | 453 (57.20) |
| Yes | 2065 (46.77) | 6558 (41.02)* | 4599 (50.24) | 11 007 (41.63)* | 154 (48.58) | 339 (42.80)§ |

Musculoskeletal pain=muscle/joint pain.
*p<0.001
†p>0.05 or not significant (NS).
‡State composition of geopolitical zones in Nigeria: south-west (Ekiti, Lagos, Ogun, Ondo, Osun and Oyo); south-south (Akwa-Ibom, Bayelsa, Cross-River, Rivers, Delta and Edo); south-east (Abia, Anambra, Ebonyi, Enugu and Imo); north-central (Benue, Kogi, Kwara, Nasarawa, Niger and Plateau States, as well as the Federal Capital Territory); north-west (Jigawa, Kaduna, Kano, Katsina, Kebbi, Sokoto and Zamfara); and north-east (Adamawa, Bauchi, Borno, Gombe, Taraba and Yobe).
§p<0.05.
GIT, gastrointestinal (nausea+vomiting).

## DISCUSSION
### Principal findings
In this study, we developed and validated symptom's prediction scores for COVID-19 positivity, independently in children, adults and the elderly patients in the Nigerian context. The best individual symptom predictors of COVID-19 positivity in children, adult and the elderly patients were loss of smell (AUROC 0.56, 95% CI 0.55 to 0.56), either fever or cough (AUROC 0.57, 95% CI 0.56 to 0.58) and difficulty in breathing (AUROC 0.53, 0.48 to 0.58), respectively. In adults, all the symptom scores showed similar performance, with the statistically weighted score (AUROC 0.65) slightly showing better performance than the unweighted (AUROC 0.63) and clinically derived weighted (AUROC 0.64) scores. Similar results were found in children and elderly patients. Overall, none of the symptom scores had good enough discrimination to use in practice.

### Strengths and limitations of this study
To the best of our knowledge, this is the first study to have developed and validated symptom prediction scores with a view to aiding prompt recognition of COVID-19 by frontline healthcare workers in the Nigerian context and possibly in sub-Saharan Africa at large. Despite the limited accuracy of the developed prediction tool, the findings are very important for a country with limited capacity for molecular diagnosis of COVID-19, as it provides the evidential basis for advocacy for more investments in molecular diagnostics by policy makers in Nigeria. We also adopted a transparent methodology and adhered to the TRIPOD reporting statement, hence minimising the vagueness often associated with the reporting of studies on the predictive performance of diagnostic models for COVID-19.[9] The methodology taken to the derivation of

weighted scores is also a strength of this study. In accordance with the preferred approach for building prediction models,[9] the participatory approach taken to deriving the clinically weighted score can enhance the study relevance in the medical community.[22] The use of beta regression coefficients as opposed to ORs in deriving the statistical weighted scores has the advantage of being less prone to bias by small to moderate sample size.[23] A common limitation of many COVID-19 diagnostic models is bias due to overfitting of the models on data that are not representative of the target population.[9] By using SORMAS database (hosts data from all over the country) for both derivation and validation in the present study, our findings are considerably generalisable to the COVID-19 situation in Nigeria and less prone to overestimation of COVID-19 risk among individuals tested.

This study, however, has some limitations that warrant discussion. First, being an analysis of secondary data based on practical recording of routine clinical assessments, the fundamental assumption is that the data recorded on clinical symptoms are reasonably complete; for instance, we assumed that where a symptom was not recorded as being absent rather than missing. Without any objective means of verifying this assumption, any bias caused by misclassification of the individual symptoms could potentially minimise differences in comparisons, in which case observed differences are likely to be in the direction of the null hypothesis. Second, the approach of splitting the dataset for both derivation and validation of symptom scores may have lowered the precision of estimated effect (wider 95% CIs)[24] and potentially underestimated prediction performance due to loss of power.[25] Moreover, evidence supporting the 10 events per outcome rule of thumb has been found by van Smeden *et al*[26] to be weak. Third, the study lacked detailed clinical laboratory data, such as record for albumin or albumin/globin, direct bilirubin values and red cell distribution width, which have been found to be significant variables in COVID-19 diagnostic models.[9] Practically, however, the time and technical

**Table 2** Associations and predictive performance of individual clinical characteristics of the study participants in relation to COVID-19 positivity (based on derivation datasets)

| Variable | Children (n=7911) | | | Adults (n=13 167) | | | Elderly (n=399) | | |
|---|---|---|---|---|---|---|---|---|---|
| | Unadjusted OR (95% CI) | Adjusted OR (95% CI)* | AUROC (95% CI)† | Unadjusted OR (95% CI) | Adjusted OR (95% CI)‡ | AUROC (95% CI)† | Unadjusted OR (95% CI) | Adjusted OR (95% CI)‡ | AUROC (95% CI)† |
| **Obvious (visible to healthcare workers on sight)** | | | | | | | | | |
| Chills/sweat | | | | | | | | | |
| No | 1.00 | 1.00 | | 1.00 | 1.00 | | 1.00 | 1.00 | |
| Yes | 1.15 (0.77 to 1.70)NS | | | 1.14 (0.85 to 1.54)NS | | | 0.30 (0.03 to 2.57)NS | | |
| Cough | | | | | | | | | |
| No | 1.00 | 1.00 | 0.54 | 1.00 | 1.00 | 0.57 | 1.00 | 1.00 | 0.52 |
| Yes | 1.39 (1.26 to 1.53)‡ | 1.32 (1.19 to 1.47)‡ | (0.53 to 0.55) | 1.85 (1.72 to 1.99)‡ | 1.77 (1.64 to 1.91)‡ | (0.56 to 0.58) | 1.61 (1.06 to 2.44)§ | 1.59 (1.04 to 2.43)§ | (0.47 to 0.57) |
| Breathing difficulty | | | | | | | | | |
| No | 1.00 | | | 1.00 | 1.00 | | 1.00 | 1.00 | 0.53 |
| Yes | 1.02 (0.87 to 1.19)NS | | | 1.23 (1.11 to 1.35)‡ | 1.10 (0.99 to 1.22)NS | | 1.71 (1.11 to 2.66)§ | 1.74 (1.12 to 2.72)§ | (0.48 to 0.58) |
| Rapid breathing | | | | | | | | | |
| No | 1.00 | 1.00 | | 1.00 | 1.00 | | 1.00 | | |
| Yes | 1.11 (0.73 to 1.68)NS | | | 1.58 (1.15 to 2.16)§ | 1.24 (0.89 to 1.73)NS | | 2.04 (0.45 to 9.23)NS | | |
| Runny nose | | | | | | | | | |
| No | 1.00 | 1.00 | 0.52 | 1.00 | 1.00 | 0.53 | 1.00 | | |
| Yes | 1.53 (1.37 to 1.70)‡ | 1.48 (1.32 to 1.66)‡ | (0.51 to 0.53) | 1.30 (1.20 to 1.42)‡ | 1.18 (1.08 to 1.30)‡ | (0.52 to 0.54) | 1.12 (0.66 to 1.88)NS | | |
| **Elicitable (can be found out from patients' responses or by asking questions of relatives)** | | | | | | | | | |
| Abdominal pain/diarrhoea | | | | | | | | | |
| No | 1.00 | | | 1.00 | 1.00 | | 1.00 | | |
| Yes | 0.87 (0.71 to 1.07)NS | | | 1.12 (0.98 to 1.29)NS | | | 0.62 (0.26 to 1.44)NS | | |
| GIT symptoms | | | | | | | | | |
| No | 1.00 | | | 1.00 | 1.00 | | 1.00 | | |
| Yes | 0.96 (0.82 to 1.14)NS | | | 1.17(1.04 to 1.32)§ | 1.11 (0.98 to 1.26)NS | | 0.91 (0.45 to 1.83)NS | | |
| Chest pain | | | | | | | | | |
| No | 1.00 | | | 1.00 | 1.00 | 0.51 | 1.00 | | |
| Yes | 1.33 (0.98 to 1.79)NS | | | 1.75 (1.44 to 2.14)‡ | 1.50 (1.21 to 1.85)‡ | (0.50–0.51) | 1.53 (0.48 to 4.83)NS | | |
| Fatigue/weakness | | | | | | | | | |
| No | 1.00 | 1.00 | 0.51 | 1.00 | 1.00 | 0.51 | 1.00 | | |
| Yes | 1.29 (1.03 to 1.62)§ | 1.53 (1.21 to 1.93)‡ | (0.51–0.52) | 1.25 (1.07 to 1.46)§ | 1.39 (1.17 to 1.64)‡ | (0.50 to 0.51) | 2.00 (0.90 to 4.35)NS | | |
| Headache | | | | | | | | | |
| No | 1.00 | | | 1.00 | 1.00 | 0.52 | 1.00 | | |
| Yes | 1.12 (0.98 to 1.29)NS | | | 1.44 (1.29 to 1.60)‡ | 1.24 (1.11 to 1.39)‡ | (0.51 to 0.52) | 0.67 (0.31 to 1.44)NS | | |
| Musculoskeletal pain | | | | | | | | | |

Continued

**Table 2** Continued

| Variable | Children (n=7911) | | | Adults (n=13 167) | | | Elderly (n=399) | | |
|---|---|---|---|---|---|---|---|---|---|
| | Unadjusted OR (95% CI) | Adjusted OR (95% CI)* | AUROC (95% CI)† | Unadjusted OR (95% CI) | Adjusted OR (95% CI)‡ | AUROC (95% CI)† | Unadjusted OR (95% CI) | Adjusted OR (95% CI)‡ | AUROC (95% CI)† |
| No | 1.00 | | | 1.00 | | | 1.00 | | |
| Yes | 1.09 (0.80 to 1.48)NS | | | 1.04 (0.81 to 1.32)NS | | | 1.80 (0.59 to 5.45)NS | | |
| Sore throat | | | | | | | | | |
| No | 1.00 | | | 1.00 | | | 1.00 | | |
| Yes | 1.02 (0.91 to 1.14)NS | | | 0.83 (0.76 to 0.90)‡ | | | 0.82 (0.50 to 1.35)NS | | |
| Loss of taste | | | | | | | | | |
| No | 1.00 | 1.00 | 0.54 | 1.00 | 1.00 | 0.54 | 1.00 | | |
| Yes | 7.07 (5.59 to 8.95) | **2.26 (1.63 to 3.12)‡** | (0.54 to 0.55) | 5.73 (4.75 to 6.91)‡ | **2.33 (1.83 to 2.96)‡** | (0.53 to 0.54) | 4.67 (0.93 to 23.42)NS | | |
| Loss of smell | | | | | | | | | |
| No | 1.00 | 1.00 | **0.56** | 1.00 | 1.00 | 0.55 | 1.00 | 1.00 | 0.51 |
| Yes | 7.85 (6.30 to 9.77)‡ | **4.87 (3.66 to 6.49)‡** | **(0.55 to 0.56)** | 6.74 (5.58 to 8.14)‡ | **4.18 (3.30 to 5.28)‡** | (0.54 to 0.55) | 5.48 (1.12 to 26.72)§ | **7.15 (1.44 35.44)§** | (0.49 to 0.52) |
| **Measurable** Fever | | | | | | | | | |
| No | 1.00 | 1.00 | 0.54 | 1.00 | 1.00 | **0.57** | 1.00 | | |
| Yes | 1.43 (1.29 to 1.58)‡ | **1.47 (1.33 to 1.63)‡** | (0.52 to 0.55) | 1.64 (1.52 to 1.76)‡ | **1.58 (1.47 to 1.71)‡** | **(0.56 to 0.58)** | 1.42 (0.95 to 2.12)NS | | |

Significant results in the adjusted model are in bold fonts.
*The fully adjusted model includes all statistically significant variables from the unadjusted model.
†Area under receiver operating characteristic curve; figures are for variables that were statistically significant in the adjusted model only; the best predictive AUROC value is highlighted in bold.
‡Wald's p values: <0.001.
§Wald's p values: <0.05.
NS, not significant (p>0.05).;

**Table 3** Predictive performance of unweighted, statistically and clinically weighted score thresholds for predicting COVID-19 positivity in children, adults and elderly

| Outcome | Score | Unweighted score AUROC value (95% CI) | Statically weighted score | Clinically weighted score |
|---|---|---|---|---|
| **Children (<17 years), n=8077** | | | | |
| Combined score | | 0.6064 | 0.6177 | 0.5915 |
| Symptom threshold | ≥1 | 0.55 (0.54 to 0.56) | 0.55 (0.54 to 0.56) | 0.55 (0.54 to 0.56) |
| | ≥2 | **0.58 (0.57 to 0.59)** | 0.55 (0.54 to 0.56) | 0.54 (0.53 to 0.55) |
| | ≥3 | 0.55 (0.54 to 0.56) | 0.55 (0.54 to 0.57) | 0.56 (0.55 to 0.57) |
| | ≥4 | 0.52 (0.52 to 0.53) | **0.59 (0.58 to 0.60)** | **0.57 (0.56 to 0.58)** |
| **Adults (17–64 years), n=13 274** | | | | |
| Combined score | | 0.6333 | 0.6475 | 0.6389 |
| Symptom threshold | ≥1 | 0.56 (0.55 to 0.56) | 0.56 (0.55 to 0.56) | 0.56 (0.55 to 0.56) |
| | ≥2 | **0.61 (0.60 to 0.61)** | 0.58 (0.57 to 0.58) | 0.56 (0.55 to 0.56) |
| | ≥3 | 0.56 (0.55 to 0.57) | 0.58 (0.57 to 0.59) | **0.58 (0.58 to 0.59)** |
| | ≥4 | 0.53 (0.53 to 0.54) | **0.59 (0.58 to 0.59)** | **0.58 (0.58 to 0.59)** |
| **Elderly (≥65 years), n=393** | | | | |
| Combined score | | 0.5413 | 0.5453 | 0.5426 |
| Symptom threshold | ≥1 | 0.52 (0.48 to 0.57) | 0.52 (0.48 to 0.57) | 0.52 (0.48 to 0.57) |
| | ≥2 | **0.53 (0.49 to 0.57)** | 0.52 (0.48 to 0.57) | 0.52 (0.48 to 0.57) |

The best predictive AUROC values for each age group are highlighted in bold.
AUROC, area under the receiver operating characteristic curve.

requirements for testing these laboratory data could limit their clinical utility.

### Interpretation and implications of findings

Based on the systematic and critical review of diagnostic scores by Wynants *et al*, the performance of a diagnostic model is influenced by its composition, with higher number of clinical and laboratory parameters in a model indicating better predictive performance.[9] For instance, studies in China,[27 28] Brazil,[29] Italy,[30] The Netherlands[31] and France[32] with several clinical and laboratory parameters recorded excellent discriminatory performance, although with substantial evidence of bias[9] and limited clinical utility. Conversely, a prediction model (containing fewer number of symptoms, heart rate, systolic and diastolic blood pressure) developed by Sun *et al*[33] in Singapore had a poor discriminatory capacity (C statistic: 0.65; 0.57–0.73). There is evidence to further suggest that the discriminatory accuracy of a prediction model, particularly its sensitivity, can be enhanced by including certain variables including a combination of loss of smell or taste

and fever.[34] In the absence of comparison study from a sub-Saharan African country, it is difficult to fully explain the variation in the findings from the present study and elsewhere. Thus, a follow-up study using both clinical and laboratory parameters in a Nigerian setting or in sub-Saharan Africa (with similar healthcare system and demographic structure) is recommended.

Prediction performance of the unweighted score with regard to COVID-19 positivity was better in adults than in children and the elderly patients in our study, although the predictive capacity of all the scores was poor overall. This finding has an important implication on NCDC's current definition of COVID-19 suspected cases, which emphasises acute respiratory symptoms and either travel history within 14 days prior to symptom onset or self-reported contact with a confirmed case.[13] Given our findings are indicative of age dependency of symptom, it may be useful to review the current case definitions of COVID-19 in Nigeria. For example, we found loss of smell and either fever or cough to be better in

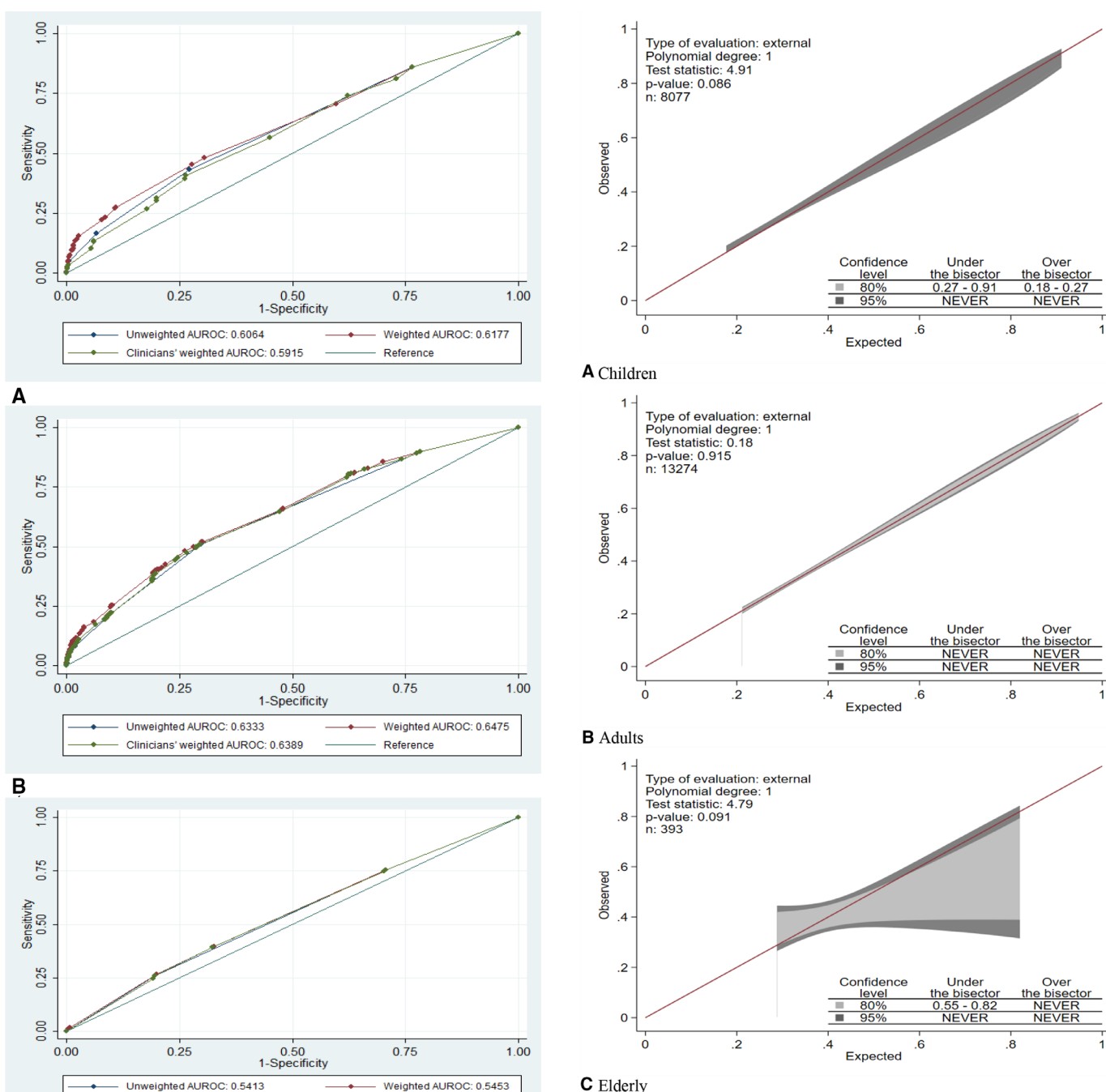

**Figure 2** Comparison of unweighted, statistically weighted and clinically weighted scores for predicting COVID-19 positivity in children (A), adults (B) and elderly (C).

**Figure 3** Calibration belts and tests for children (A), adults (B) and elderly (C).

predicting COVID-19 positivity in children and adults, respectively, while breathing difficulty was more predictive of the disease in the elderly patients. Furthermore, this finding potentially has implications on the clinical utility of existing suspected case definition in Nigeria[13] with a high proportion of asymptomatic COVID-19 cases[8] and testing system that allows persons who are concerned about their COVID-19 risk to be tested. Thus, to minimise missed diagnoses and overburdening of the healthcare

system, with attendant psychological effects on health personnel,[35] there is a need for more economic investments on molecular testing across Nigeria.

Loss of smell recorded the highest specificity with regard to COVID-19 positivity for the three age groups: 98.1% in children, 98.5% in adults and 99.1% in the elderly. However, unlike the present study which explored the predictive capacity of loss of smell and taste separately, a combination of both symptoms has been shown to be more predictive.[36] Thus, the potential use of both loss of smell and taste to differentiate COVID-19 from endemic febrile and respiratory illnesses in Nigeria, such as malaria

and pneumonia, with overlapping symptoms warrant further study. Additionally, possibility of using both loss of smell and taste as early indicators of emerging COVID-19 wave or a surge in Nigeria would be useful in improving COVID-19 response, such as allocation of already limited testing resources, risk communication and aid decision-making concerning lockdowns and quarantines.[37] The poor predictive capacity of cough or fever alone in the present study is congruent with that in a meta-analysis.[38]

Clinical validity (characterised by sensitivity, specificity and AUROC values) is an important criterion for assessing a clinical prediction tool[39] as it is—the ability of the prediction tool to distinguish between who has an outcome (in this case SARS-CoV-2 infection) and who does not.[40] The clinical validity of all our prediction scores was generally poor but appeared to be dependent on the number of symptoms. For instance, in our study, the unweighted and weighted (both statistical and clinical) predictive scores presenting with fewer number of symptoms were more sensitive compared with many symptoms in children and adults; it was, however, the opposite relative to specificity given ≥4 symptoms recorded higher specificity values than lower symptom thresholds. The poor sensitivity of many symptoms could potentially be attributable to a high proportion of false negatives, suggesting that some symptoms have limited validity for COVID-19 in children and adults. However, similarity in the predictive performance of various symptom thresholds on the two weighted scores in elderly suggests that weighting has less predictive value for this group of population. The high specificity of more symptoms could be indicative of low proportion of false positives, underlining the need to accurately assess symptoms. In practice, there is a trade-off between sensitivity and specificity such that when the consequences of having a false positive test is very serious, specificity is prioritised over sensitivity and vice versa.[41] This is the case for the various symptom thresholds on the unweighted scale where specificity is higher than sensitivity. A higher specificity over sensitivity is of practical relevance when the political implication of refusing to test someone with suspected COVID-19 is considered, although higher sensitivity over specificity might be given preference in the early phases of a pandemic before surge capacity is reached.

Given the rapid increase in community transmission of COVID-19 cases and deleterious impacts of instituting another lockdown (partial or complete), large-scale surveillance for capturing the epidemiological trend of COVID-19 in Nigeria is crucial. However, Nigeria has limited SARS-CoV-2 testing capacity with an average turn-around of 2 days, making syndromic surveillance (symptomatic monitoring) a viable complementary surveillance system. As such, our findings would be relevant in informing the design of such a surveillance system, which has been demonstrated in Japan[42 43] and in the USA,[44] to be useful in improving the understanding of COVID-19 epidemiology (often in real time), assessing the effectiveness of public health interventions and enhancing

preparedness for the emergence of COVID-19 wave or a surge. For instance, an evaluation of a syndromic surveillance system in the USA found new taste/smell loss to be highly correlated with a range of COVID-19 outcomes, highlighting their usefulness in supporting the surveillance system as an early warning system for COVID-19 prevention and control. However, the feasibility (eg, considering selection bias and recall bias) and acceptability of a syndromic surveillance system first need to be ascertained given the large proportion of asymptomatic COVID-19 cases at diagnosis in Nigeria.[8] PPVs across the various prediction thresholds, especially for the weighted scales, were generally low despite increasing proportionately with the thresholds. This could be attributable, in part, to the general mildness of the pandemic with resultant low incidence of mortality in Nigeria. For instance, 66% of the 12 289 confirmed COVID-19 cases in Nigeria between 27 February and 6 June 2020 were asymptomatic at diagnosis, with an overall cumulative incidence and case fatality rate of 5.6 per 100 000 population and 2.8%, respectively[8]—these figures were substantially lower than those from European countries during the same period.[45] As such, our predictive tools could perform differently during a more severe COVID-19 outbreak in Nigeria.

## CONCLUSION

This study has investigated the possibility of using symptoms to predict COVID-19 positivity in Nigeria and found the predictive capacity of various symptom scores to be poor overall. However, the findings have the potential to serve as an advocacy tool for more investments in resources for capacity strengthening of molecular testing for COVID-19 in Nigeria, which is crucial for improving both clinical case management and surveillance.

**Author affiliations**
[1]Nigeria Centre for Disease Control, Abuja, Nigeria
[2]Department of Global Public Health, Karolinska Institutet, Stockholm, Sweden
[3]Department of Epidemiology and Public Health, University of Nottingham, Nottingham, UK
[4]World Health Organization, Abuja, Nigeria
[5]University of Abuja Teaching Hospital, Gwagwalada, Nigeria
[6]Lagos State Ministry of Health, Ikeja, Nigeria
[7]Department of Epidemiology and Community Health, University of Ilorin Teaching Hospital, Ilorin, Nigeria
[8]Department of Internal Medicine, National Hospital, Abuja, Nigeria
[9]National Agency for the Control of AIDS, Abuja, Nigeria
[10]Federal Medical Centre Asaba, Asaba, Nigeria
[11]Federal Teaching Hospital Gombe, Gombe, Nigeria
[12]Kaduna State Infectious Disease Control Center Community Medicine, Kaduna, Nigeria
[13]Mainland Hospital Yaba, Lagos, Nigeria
[14]Ministry of Health, Lagos, Nigeria
[15]Lagos State Government Ministry of Health, Ikeja, Nigeria
[16]Department of Community Medicine, Ahmadu Bello University, Zaria, Nigeria
[17]Department of Community Medicine, Nnamdi Azikiwe University Teaching Hospital, Nnewi, Nigeria
[18]Department of Medical Education, Usmanu Danfodiyo University, Sokoto, Nigeria
[19]Department of Community Medicine, University College Hospital, Ibadan, Nigeria
[20]School of Public Health, University of the Western Cape, Bellville, South Africa

**Acknowledgements** We acknowledge the contribution of all the COVID-19 treatment centres and laboratories within the NCDC Laboratory Networks of both epidemiological and laboratory data to SORMAS on a regular basis. We express our profound gratitude to the NCDC SORMAS team for collating and maintaining a quality dataset utilised for this study. Last but not least, we thank Professor Jo Leonard-Bee of the University of Nottingham for her input on statistical analyses.

**Contributors** KOE contributed to the conceptualisation, literature search, data management, data analysis, writing (original draft, review and editing) and formatting of the document. OA contributed to the conceptualisation, writing (review and editing), formatting of manuscript and funding acquisition. JG contributed to the conceptualisation, data management, data analysis, writing (original draft, review and editing) and formatting of the document. PM contributed to the writing (original draft, review and editing), literature search and formatting of the manuscript. CLO contributed to the writing (review and editing) and formatting of the manuscript. CK contributed to the writing (review and editing) and formatting of the manuscript. TO contributed to the writing (review and editing) and formatting of the manuscript. GG contributed to the writing (review and editing) and formatting of the manuscript. AO contributed to the writing (review and editing) and formatting of the manuscript. HGZ contributed to the writing (review and editing) and formatting of the manuscript. CO contributed to the writing (review and editing) and formatting of the manuscript. BE contributed to the writing (review and editing) and formatting of the manuscript. OI contributed to the writing (review and editing) and formatting of the manuscript. CD-N contributed to the writing (review and editing) and formatting of the manuscript. WN contributed to the writing (review and editing) and formatting of the manuscript. IAA contributed to data acquisition and curation, and writing (review) of the manuscript. OWA contributed to the writing (review and editing) and formatting of the manuscript. OF contributed to the writing (review and editing) and formatting of the manuscript. CA contributed to data acquisition and curation, and writing (review) of the manuscript. CE contributed to the writing (review and editing) and formatting of the manuscript. A Jafiya contributed to the writing (review and editing) and formatting of the manuscript. AO contributed to the writing (review and editing) and formatting of the manuscript. JTA contributed to the writing (review and editing) and formatting of the manuscript. EJ contributed to the writing (review and editing) and formatting of the manuscript. AB contributed to the writing (review and editing) and formatting of the manuscript. BM contributed to the writing (review and editing) and formatting of the manuscript. BS contributed to the writing (review and editing) and formatting of the manuscript. A Jinadu contributed to the writing (review and editing) and formatting of the manuscript. KH contributed to the writing (review and editing) and formatting of the manuscript. C Ibeh contributed to the writing (review and editing) and formatting of the manuscript. SB contributed to the writing (review and editing) and formatting of the manuscript. MCA contributed to the writing (review and editing) and formatting of the manuscript. NM contributed to supervision, writing (review), funding acquisition, project administration and resources. JO contributed to supervision, writing (review), funding acquisition, project administration and resources. EI contributed to supervision, writing (review), funding acquisition, project administration and resources. TA contributed to the writing (review and editing) and formatting of the manuscript. EI contributed to the writing (original draft, review and editing), literature search and formatting of the manuscript. C Ihekweazu contributed to conceptualisation, supervision, writing (review and editing), funding acquisition, project administration and resources. All authors invest significant contributions and approved the final draft.

**Funding** The authors have not declared a specific grant for this research from any funding agency in the public, commercial or not-for-profit sectors.

**Competing interests** None declared.

**Patient consent for publication** Not required.

**Ethics approval** The Nigeria National Health Research Ethics Committee approved the use of the SORMAS database for this study (reference number: NHREC/01/01/2 007-22/06/2020).

**Provenance and peer review** Not commissioned; externally peer reviewed.

**Data availability statement** Data are available upon reasonable request. The dataset utilised for this study can be made available upon reasonable request to the Head of NCDC Research at chinwe.ochu@ncdc.gov.ng.

**ORCID iDs**
Kelly Osezele Elimian http://orcid.org/0000-0003-2729-1613
Chinwe Lucia Ochu http://orcid.org/0000-0002-0630-7332
Giulia Gaudenzi http://orcid.org/0000-0003-4923-6965
Oluwatosin Wuraola Akande http://orcid.org/0000-0001-6906-895X

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
