## [Reviewer comments · BMJ Open]

ARTICLE DETAILS

TITLE (PROVISIONAL)	Assessing the capacity of symptom scores to predict COVID-19 positivity in Nigeria: a national derivation and validation cohort study
AUTHORS	Elimian, Kelly; Aderinola, Olaolu; Gibson, Jack; Myles, Puja; Ochu, Chinwe; King, Carina; Okwor, Tochi; Gaudenzi, Giulia; Olayinka, Adebola; Zaiyad, Habib; Ohonsi, Cornelius; Ebhodaghe, Blessing; Dan-Nwafor, Chioma; Nwachukwu, William; Abdus-salam, Ismail; Akande, Oluwatosin; Falodun, Olanrewaju; Arinze, Chinedu; Ezeokafor, Chidiebere; Jafiya, Abubakar; Ojimba, Anastacia; Aremu, John; Joseph, Emmanuel; Bowale, Abimbola; Mutiu, Bamidele; Saka, Babatunde; Jinadu, Arisekola; Hamza, Khadeejah; Ibeh, Christian; Bello, Shaibu; Asuzu, Michael; Mba, Nwando; Oladejo, John; Ilori, Elsie; Alfvén, Tobias; Igumbor, Ehimario; Ihekweazu, Chikwe

VERSION 1 – REVIEW

REVIEWER	Wang, Cheng The First Affiliated Hospital of University of Science and Technology of China
REVIEW RETURNED	23-Mar-2021

GENERAL COMMENTS	Dear editors and authors, The paper was very insightful; it's my honor to have a chance to review this kind of large sample size paper. The follows was my comments. The AUC was very small, and the results were not ideal enough, would you please further explain in the discussion part, what's the novelty and meaning of this paper? Subgroup analysis may be more suitable for present the association of clinical characteristics in different age group, if you can combine the table2/3/4 to a one table in subgroup, which may simplify and easier for readers to compare the association of clinical characteristics. The Hosmer-Lemeshow goodness-of-fit tests showed a good fitness of the model, the process of statistical analysis was well done, the authors had make a big effort to clarify the associations and to predict the capacity. However, I cannot see a calibration curve of the predicted probability and observed probability in the paper, which was not accuracy enough. The datasets was randomly split in two equal halves as derivation and validation parts; however, the authors did not clarify the methods of randomly splitting. Data is one of a common way for splitting.
---

	The results was negative, please further clarify the conclusion, the sentence like this is not enough, "This emphasises the need to continue the capacity strengthening of molecular laboratory to determine COVID-19 positivity in Nigeria. Because, which is always a common sense around the word, which did not embody the novelty of this study. In the limitation part, I believe there are still more than two limitations as you stated of these study.
--	---

REVIEWER	Desjardins, M.R. Johns Hopkins University Bloomberg School of Public Health, Epidemiology
REVIEW RETURNED	29-Apr-2021

GENERAL COMMENTS	I commend the authors for this great manuscript. It is extremely well-written and the statistical analyses are appropriate and clearly described, with the exception of a few limitations. I strongly recommend this study for publication once my comments are addressed. (1) Missing some literature on syndromic surveillance of COVID-19. Please review and cite the following: Desjardins, M. R. (2020). Syndromic surveillance of COVID-19 using crowdsourced data. The Lancet Regional Health–Western Pacific, 4. Güemes, A., Ray, S., Aboumerhi, K., Desjardins, M. R., Kvit, A., Corrigan, A. E., ... & Etienne-Cummings, R. (2021). A syndromic surveillance tool to detect anomalous clusters of COVID-19 symptoms in the United States. Scientific reports, 11(1), 1-11. Yoneoka, D., Tanoue, Y., Kawashima, T., Nomura, S., Shi, S., Eguchi, A., ... & Miyata, H. (2020). Large-scale epidemiological monitoring of the COVID-19 epidemic in Tokyo. The Lancet Regional Health-Western Pacific, 3, 100016. Nomura, S., Yoneoka, D., Shi, S., Tanoue, Y., Kawashima, T., Eguchi, A., ... & Miyata, H. (2020). An assessment of self-reported COVID-19 related symptoms of 227,898 users of a social networking service in Japan: Has the regional risk changed after the declaration of the state of emergency?. The Lancet Regional Health-Western Pacific, 1, 100011. (2) Did you consider adding the socioeconomic variables in your models? I think that targeted interventions can greatly benefit from examining the effects of geopolitical zones, urban/rural status, education, and others. (3) I have a slight issue with the model validation. I believe that you did lose power by splitting the dataset in half. I think a cross-validation approach (e.g. k-fold, repeated k-fold, etc.) would have been more appropriate and reduce bias & uncertainty of the results. Please comment.
--

VERSION 1 – AUTHOR RESPONSE

Reviewer: 1

Dr. Cheng Wang, The First Affiliated Hospital of University of Science and Technology of China
The paper was very insightful; it's my honour to have a chance to review this kind of large sample size paper. The follows were my comments.

R: We are grateful for your overall assessment and individual review of the paper.

1. The AUC was very small, and the results were not ideal enough, would you please further explain in the discussion part, what's the novelty and meaning of this paper?

R: Thank you for your comments. We agree with the view that the AUC was small and that the results, compared to some previously published papers, could be considered uninteresting. However, this is the point we are trying to make in the paper by demonstrating that reliance on clinical signs and symptoms alone for diagnosing COVID-19 is not a sensible strategy. This is important evidence that provides the justification for increased investment in molecular testing capacity in the Nigerian context.

Regarding novelty, we had noted in the strengths of the study section of the discussion chapter that to the best of our knowledge, this is the first manuscript from Nigeria (and possibly from Sub-Saharan Africa), focused on the development and validation of clinical scoring tools for COVID19, hence contributing to the global literature. Given the different underlying infectious disease epidemiology of this region, with endemic malaria and Lassa fever, it's reasonable to expect poorer performance of such tools and this is an important finding.

We have now clarified in the paper how the generated evidence could potentially be useful in serving as an advocacy tool for more investments in Nigeria's diagnostics for COVID-19. For a country like Nigeria where molecular diagnosis of infectious diseases (including COVID-19) is underfunded, we strongly believe that this paper would be useful to public health agencies in making a strong case for more investments.

2. Subgroup analysis may be more suitable to present the association of clinical characteristics in different age group, if you can combine the table 2/3/4 to a one table in subgroup, which may simplify and easier for readers to compare the association of clinical characteristics.

R: Thank you very much for the suggestion, which we fully agreed with. We have now combined Tables 2, 3, and 4 into a single Table (i.e. Table 2), making the results among the three age groups easier to compare and more simplified. The numbering of all the Tables has also changed accordingly (from 5 to 3 Tables).

3. The Hosmer-Lemeshow goodness-of-fit tests showed a good fitness of the model, the process of statistical analysis was well done, the authors had made a big effort to clarify the associations and to predict the capacity. However, I cannot see a calibration curve of the predicted probability and observed probability in the paper, which was not accurate enough.

R: Thank you for your comments and the recommendation to perform a calibration curve, they are well appreciated. We have used the approach recommended by Nattino and colleagues (<https://ideas.repec.org/a/tsj/stataj/v17y2017i4p1003-1014.html>) to provide calibration belt and test, separately for children, adults and elderly patients. We have created separate calibration belt and tests for each score (unweighted, statistically- and clinically-weighted) in children, adults, and elderly, but presented in a Supplementary Figure. Lastly, we have deleted the Hosmer-Lemeshow goodness-of-fit and its corresponding supplementary Table 12 from the revised manuscript.

4. The dataset was randomly split into two equal halves as derivation and validation parts; however, the authors did not clarify the methods of randomly splitting. Data is one of the common ways for splitting.

R: Thank you for pointing out the need for being explicit in specifying the splitting method used in the paper. In lines 177-179, we have tried to be more explicit by specifying the splitting method used as follow: “We used the train-test split-sample method to randomly split the complete dataset into two (i.e. training and testing datasets), assigning half of the records to either the training (derivation) or testing (validation) datasets based on a random number generated within the statistical software.”

5. The results was [were] negative, please further clarify the conclusion, the sentence like this is not enough, “This emphasises the need to continue the capacity strengthening of the molecular laboratory to determine COVID-19 positivity in Nigeria. Because, which is always a common sense around the world, which did not embody the novelty of this study.

R: Thank you for this observation, considering the importance of the conclusion section of the paper. While highlighting the poor diagnostic capacity of the tool, we have tried to capture the novelty of the study, both in the strengths section of the discussion chapter and in the conclusion. Essentially, we have now reiterated the potential of the findings from the paper to serve as an advocacy tool for more investment of resources for capacity strengthening of molecular testing for COVID-19 in Nigeria. Moreover, we think that having this paper published in a peer-reviewed journal, such as the BMJ Open, would add credence to such a case.

6. In the limitation part, I believe there are still more than two limitations as you stated in this study.

R: Thank you for your comments; addressing them would help improve the paper’s quality overall. Nonetheless, we had actually outlined three major limitations of this paper rather than two. The first limitation focused on the potential misclassification of symptoms based on the fundamental assumption made; the second focused on the potential loss of power given the approach used to split the dataset; and the third focused on the dearth of information on laboratory parameters, which have been shown to be useful in developing a predictive model in previous papers. To make these points more explicit, we have now used an introductory numbering system, such as first, second, and third.

Reviewer: 2

Dr. M.R. Desjardins, Johns Hopkins University Bloomberg School of Public Health

I commend the authors for this great manuscript. It is extremely well-written and the statistical analyses are appropriate and clearly described, with the exception of a few limitations. I strongly recommend this study for publication once my comments are addressed.

R: R: We are grateful for your overall assessment and individual review of the paper, particularly for your conditional recommendation of our paper for publication.

(1) Missing some literature on syndromic surveillance of COVID-19. Please review and cite the following:

1. Desjardins, M. R. (2020). Syndromic surveillance of COVID-19 using crowdsourced data. *The Lancet Regional Health–Western Pacific*, 4.
2. Güemes, A., Ray, S., Aboumerhi, K., Desjardins, M. R., Kvit, A., Corrigan, A. E., ... & Etienne-Cummings, R. (2021). A syndromic surveillance tool to detect anomalous clusters of COVID-19 symptoms in the United States. *Scientific reports*, 11(1), 1-11.
3. Yoneoka, D., Tanoue, Y., Kawashima, T., Nomura, S., Shi, S., Eguchi, A., ... & Miyata, H. (2020). Large-scale epidemiological monitoring of the COVID-19 epidemic in Tokyo. *The Lancet Regional Health-Western Pacific*, 3, 100016.
4. Nomura, S., Yoneoka, D., Shi, S., Tanoue, Y., Kawashima, T., Eguchi, A., ... & Miyata, H. (2020). An assessment of self-reported COVID-19 related symptoms of 227,898 users of a social networking service in Japan: Has the regional risk changed after the declaration of the state of emergency? *The Lancet Regional Health-Western Pacific*, 1, 100011.

R: Thank you very much for recommending these very important and relevant articles on syndromic surveillance of COVID-19, particularly with regards to its usefulness in improving prompt response and monitoring of public health interventions. By reading and citing these articles, we feel confident that the application of our findings has been expanded beyond the original focus on clinical management. A paragraph has been dedicated to outlining the importance of our findings in exploring syndromic surveillance in a Nigerian context, while also mentioning the need to first evaluate the feasibility and acceptability of the complementary surveillance system. Additionally, we have acknowledged the effectiveness of smell/taste loss in potentially serving as early indicators of the emergence of the COVID-19 wave or a surge, as reported in one of the articles. Overall, these papers have further buttressed the relevance of our paper in a setting with limited capacity for molecular diagnostics.

(2) Did you consider adding the socioeconomic variables in your models? I think that targeted interventions can greatly benefit from examining the effects of geopolitical zones, urban/rural status, education, and others.

R: Thank you for this suggestion, we believe exploring these variables (geopolitical zone of residence, urban/rural setting, education etc.) would greatly add value to the study. However, being an analysis of a surveillance dataset rather than a research-oriented dataset, the majority of these variables had substantial missing data (Table 1). Thus, adding them to the prediction model could seriously bias our estimated predictions. However, the Nigeria CDC (custodian of the analysed dataset) and its technical partners are conducting a prospective cohort study to ascertain the effectiveness of COVID-19 vaccines in Nigeria. We could address the gap you have identified and other limitations of this paper (lack of laboratory parameters) using data from the ongoing study. Additionally, there is interesting ethical issues in this area as well. Preliminary findings from unpublished qualitative research on sexual health risk targeting scores, which used postcode (a proxy for deprivation), ethnicity and region of birth in the UK (all of which are established risks for STIs in this setting), found that patients were unhappy that findings from the study might be used to statistically inform their care.

(3) I have a slight issue with the model validation. I believe that you did lose power by splitting the dataset in half. I think a cross-validation approach (e.g. k-fold, repeated k-fold, etc.) would have been more appropriate and reduce bias & uncertainty of the results. Please comment.

R: Thank you for this important comment regarding the method used for splitting the dataset and its potential impact on statistical power. We agree that cross-validation (e.g. k-fold) is a more superior cross-validation approach than the one used in the present study. Using a k-fold cross-validation approach, for example, would require us to divide the dataset into folds (e.g. 5-10 folds) to ensure that each fold has an opportunity to be used as a validation dataset. We are however retaining the existing splitting approach for the validated dataset based on the following reasons:

First, we mentioned in the data management section that despite the absence of a formal sample size calculation our study met the standard sample size requirement of 10 outcome events per degree of freedom in prediction models (e.g., 10 binary variables in the model require 100 COVID-19 positive cases). However, while power may have been lost it should still have been sufficient to address the research question. Second, SORMAS (the Nigeria CDC electronic database) is very robust in terms of coverage and is believed to capture the largest dataset (covering 36 states and the federal capital territory of Nigeria) on COVID-19 in Nigeria. In fact, our sample size is only slightly smaller than the dataset utilised for model development and validation in the US (i.e. most of the available models have a smaller sample size than ours). Third, we have acknowledged in the limitation section of the discussion chapter the potential shortcomings associated with our approach of splitting the analysed dataset. However, in the same section, we noted that the use of beta regression coefficients instead of odds ratios in deriving statistically weighted scores is less prone to bias by small to moderate sample sizes (hence our predictive values are robust and less prone to bias).

VERSION 2 – REVIEW

REVIEWER	Wang, Cheng The First Affiliated Hospital of University of Science and Technology of China
REVIEW RETURNED	22-Jun-2021

GENERAL COMMENTS	Dear authors, After reading the answering letter carefully and reviewing your draft again, I found that you had made a big effort to polish your paper. However, there are still some minor revisions you should do in the second round review. In the conclusion of abstract part, we all know that, molecular testing for covid-19 is a golden standard for diagnosing, which is a common sense. You had done a lot of work; however, we cannot deduce an inner connection between the results and the conclusion, which may cause confuse to readers. Would you please clarify the investments status and molecular testing incidence of covid-19 in your country in introduction parts? (add some references) Would you please clarify your results meaning deeper in discussion part? Such as, in your country or region, what is the current status of molecular diagnosis of COVID-19? If the clinical needs cannot be met, how many suspected patients have not undergone molecular diagnosis. If there is a lack of molecular diagnosis and only score assessed based on the patient's symptoms, what are the shortcomings? For example, there will be missed diagnosis, and it will increase the workload of medical staff, and it will increase the psychological burden of medical staff. Therefore, the government needs to invest more economically to increase molecular diagnosis, which is of great significance, especially in African area or sub-Saharan Africa. This will increase the logistic relationship of your results and conclusion. The following reference may be useful for you. Zhang Jinlong., Fang Yunyun., Lu Zhaohui., Chen Xia., Hong Na., Wang Cheng.(2021). Lacking Communication Would Increase General Symptom Index Scores of Medical Team Members During COVID-19 Pandemic in China: A Retrospective Cohort Study. Inquiry, 58(undefined), 46958021997344. doi:10.1177/0046958021997344 For Figure2 and 3, and supplementary figure 3, please add figure legends to describe the contents in short, not only figure title, which may be easier for readers to understand.
--

REVIEWER	Desjardins, M.R. Johns Hopkins University Bloomberg School of Public Health, Epidemiology
REVIEW RETURNED	21-Jun-2021

GENERAL COMMENTS	The authors did a great job addressing my comments and I am happy to share that I have nothing further to add. I strongly recommend publication.
--

VERSION 2 – AUTHOR RESPONSE

Reviewer: 1

Dr. Cheng Wang, The First Affiliated Hospital of University of Science and Technology of China
Dear authors,

After reading the answering letter carefully and reviewing your draft again, I found that you had made a big effort to polish your paper. However, there are still some minor revisions you should do in the second round review.

R: We are grateful for your initial comments and suggestions, which greatly helped to improve the quality of the manuscript.

In the conclusion of abstract part, we all know that, molecular testing for Covid-19 is a golden standard for diagnosing, which is a common sense. You had done a lot of work; however, we cannot deduce an inner connection between the results and the conclusion, which may cause confusion to readers.

R: Thank you very much for this comment on the need for clarity in the conclusion section of the abstract. As explained in our introduction, while molecular testing is considered the gold standard for diagnosis, this is not always feasible in resource-constrained settings and policy makers need a stronger, context-specific justification for further investment. This is why we aimed to explore the possibility of using clinical predictive scores to complement molecular testing, which is suboptimal in Nigeria, especially in the early phase of the pandemic. However, we found the various clinical scores to have poor predictive capacity, hence our recommendation for the use of evidence from this study as an advocacy tool for more investments in resources for molecular testing of COVID-19 in Nigeria. We think the conclusion follows from our results and is contextualised by our introduction and therefore, we have retained this except for the minor edits in the revised draft.

Would you please clarify the investments status and molecular testing incidence of Covid-19 in your country in introduction parts? (add some references)

R: Thank you for this comment. This was addressed in the introduction section of the manuscript (lines 98-110 in the original or marked manuscript). However, we have edited the section to reflect the fact that despite the investment in molecular testing and expansion of the Nigeria CDC laboratory networks, testing in Nigeria with a population of over 200 million and community transmission of COVID-19 is still well below the Africa CDC-set target of 1% for population.

Would you please clarify your results meaning deeper in discussion part? Such as, in your country or region, what is the current status of molecular diagnosis of COVID-19? If the clinical needs cannot be met, how many suspected patients have not undergone molecular diagnosis. If there is a lack of molecular diagnosis and only score assessed based on the patient's symptoms, what are the shortcomings? For example, there will be missed diagnosis, and it will increase the workload of medical staff, and it will increase the psychological burden of medical staff. Therefore, the government needs to invest more economically to increase molecular diagnosis, which is of great significance, especially in African area or sub-Saharan Africa. This will increase the logistic relationship of your results and conclusion. The following reference may be useful for you. Zhang Jinlong., Fang Yunyun., Lu Zhaohui., Chen Xia., Hong Na., Wang Cheng (2021). Lacking Communication Would Increase General Symptom Index Scores of Medical Team Members During COVID-19 Pandemic in China: A Retrospective Cohort Study. *Inquiry*, 58(undefined), 46958021997344.
doi:10.1177/0046958021997344

R: Thank you very much for your suggestions and recommended article. We have incorporated your suggestions in the discussion section of the manuscript (lines 441-446 in the original or marked

manuscript) and cited the article by Zhang and colleagues, which has now captured the mental health and psychosocial impacts of the findings on healthcare workers. Furthermore, we agree that bringing in this point in the discussion of the findings has indeed strengthened the relationship between our results and conclusion.

For Figures 2 and 3, and supplementary figure 3, please add figure legends to describe the contents in short, not only figure title, which may be easier for readers to understand.

R: Thank you very much for your suggestion, it is very much appreciated. The journal's electronic submission system requires the addition of caption to images (figures) submitted. We have done this for all the images submitted including Figures 2 and 3 as well as supplementary Figure 3. Additionally, we have provided legends for all the Figures at the end of the manuscript (lines 693-699 in the original or marked manuscript).

Reviewer: 2

Dr. M.R. Desjardins, Johns Hopkins University Bloomberg School of Public Health

The authors did a great job addressing my comments and I am happy to share that I have nothing further to add. I strongly recommend publication.

R: We are grateful for your suggestions in your previous comments and recommendation of the manuscript for publication.